

# Protein profile changes during priming explants to embryogenic response in *Coffea canephora*: identification of the RPN12 proteasome subunit involved in the protein degradation

Evelyn A. Carrillo-Bermejo[1],[*], Ligia Brito-Argáez[1],[*],
Rosa M. Galaz-Ávalos[1], Felipe Barredo-Pool[2], Víctor M. Loyola-Vargas[1]
and Victor Aguilar-Hernández[1]

[1] Unidad de Biología Integrativa, Centro de Investigacion Científica de Yucatan (CICY), Mérida, Yucatán, Mexico
[2] Unidad de Biotecnología, Centro de Investigación Científica de Yucatán (CICY), Mérida, Yucatán, Mexico
[*] These authors contributed equally to this work.

Corresponding authors
Víctor M. Loyola-Vargas,
vmloyola@cicy.mx
Victor Aguilar-Hernández,
victor.aguilar@cicy.mx

## ABSTRACT

Plant somatic embryogenesis encompasses somatic cells switch into embryogenic cells that can later produce somatic embryos with the ability to produce plantlets. Previously, we defined *in vitro* culture settings for the somatic embryogenesis process of *Coffea canephora* that comprise adequate plantlets with auxin plus cytokinin followed by cut-leaf explant cultivation with cytokinin, producing embryos with the ability to regenerate plantlets. Here, we confirmed that cultivating cut-leaf explants with cytokinin is sufficient to promote somatic embryos proliferation and the high yield of somatic embryos in the protocol requires adequate plantlets with auxin plus cytokinin. Two-dimensional sodium dodecyl sulfate-polyacrylamide gel electrophoresis gels reveal auxin-plus cytokinin-dependent regulated proteins in plantlets with up and down abundance. Chitinase A class III, proteins involved in the metabolism and folding of proteins, photosynthesis, antioxidant activity, and chromatin organization were identified. The RPN12 protein, which is a subunit of the proteasome 26S, has an abundance that is not associated with transcript changes, suggesting post-translational regulation.

# INTRODUCTION

The plasticity of plant cells is a fascinating aspect of plant development, as they can respond to form embryos through somatic embryogenesis (SE). Despite the inherent plasticity nature of plant cells, they can respond to embryogenesis and maintain embryonic competence in embryogenic lines, which is achieved through various embryogenic cultures. Without a reproducible and efficient regeneration system, it is not possible to use totipotency easily because of the internal and external experimental settings, for instance,

endogenous plant growth regulators (PGRs) and PGRs that are provided in the medium. SE settings have been established to enable plant regeneration of different plant species and have eventually resulted in clonal plantlets (*Etienne et al., 2018*; *Guan et al., 2016*; *Loyola-Vargas, 2016*; *Maruyama & Hosoi, 2019*; *Pilarska et al., 2016*). After proliferation and full development of the somatic embryo, matured embryos display well-developed protoderms and apical and radical meristems, which further result in complete plantlets.

The transition of somatic cells to embryogenic cells is initiated by an intense reshaping of the proteome (*Aguilar-Hernández & Loyola-Vargas, 2018*; *Awada et al., 2019*; *Campos et al., 2016*; *Karami, Aghavaisi & Mahmoudi Pour, 2009*; *Quintana-Escobar et al., 2019*; *Wójcikowska & Gaj, 2017*), which involves changes in the protein abundance (*Ge et al., 2015*; *Gomez-Garay et al., 2013*; *Hou et al., 2023*; *Zhu et al., 2018*), specially during SE stages (*Gomez-Garay et al., 2013*; *Imin et al., 2005*; *Rode et al., 2012*), and post-translational modifications such as acetylation or phosphorylation (*Aroonluk, Roytrakul & Jantasuriyarat, 2019*; *Xavier et al., 2022*; *Xia et al., 2016*; *Zhang et al., 2023*). Comparative proteomic analysis has uncovered protein associated with the developmental process of SE by analyzing samples from contrasting phenotypes, including embryogenic callus (EC) *vs.* non-embryogenic callus (NEC), SE *vs.* zygotic embryogenesis (ZE), EC or NEC *vs.* SE stages, or SE stages, among other types (*Aguilar-Hernández & Loyola-Vargas, 2018*; *Heringer, Santa-Catarina & Silveira, 2018*). Relevant protein classes identified include metabolic enzymes, PGR metabolism and signaling pathways, ROS scavengers, proteolytic enzymes, molecular chaperone proteins, cytoskeleton, polyamine metabolic pathway, and transcriptional regulation, among others. Most proteome rearrangements such as subcellular relocalization, changes in protein turnover, and post-translational modifications can be attributed to changes in transcription, which are driven by chromatin modifications and miRNA-mediated gene regulation, and later, interconnected signaling cascades are responsible for controlling gene arrays expression through activation or repression (*Castander-Olarieta et al., 2020*; *Wójcik, Wójcikowska & Gaj, 2020*; *Wojcikowska, Wojcik & Gaj, 2020*). The coordinated regulation of downstream target genes by transcription factors such as AGL15, LEC2, LEC1, BBM, MAD-box, and WUS has a significant impact on the embryogenic reprogramming of somatic cells and the overall developmental process of SE (*Cao et al., 2017*; *Horstman et al., 2017*; *Indoliya et al., 2016*; *Jamaluddin, Mohd Noor & Goh, 2017*; *Magnani et al., 2017*; *Yang et al., 2012*).

The reprogramming of cells can be impacted by PGRs, with mechanical damage, cocultivation of bacteria with explants, or nanoparticles application to explants, among other factors (*Arruda, da Silva & Kato, 2023*; *Fehér, 2015*; *Gorpenchenko et al., 2006*; *Murthy et al., 1999*; *Priyono et al., 2010*; *Zimmerman, 1993*). Particularly influential signals for reprogramming are the PGRs auxin and cytokinin, which drive changes in transcription (*Quintana-Escobar et al., 2023*, *2019*; *Salaün, Lepiniec & Dubreucq, 2021*; *Salvo et al., 2014*). The cytokinin-responsive genes *A-ARR* (*Arabidopsis* response regulator) type lead to the negative regulation of cytokinin signaling and somatic embryo proliferation, while the B-ARR type and DNA-binding auxin response factor (ARF) promote the somatic embryo proliferation (*Su et al., 2014*; *Wójcikowska & Gaj, 2017*). Besides precise biosynthesis, accumulation and distribution of auxin and cytokinin
(*Correia et al., 2012*; *Hatanaka et al., 1991*; *Kumaravel et al., 2017*; *Li et al., 2022*; *Márquez-López et al., 2018*; *Michalczuk, Cooke & Cohen, 1992*; *Zhou et al., 2016*), protein biosynthesis activation, post-translational modifications of proteins or the removal of unnecessary proteins may be required for the reprogramming of the cell to achieve SE.

Auxins and cytokinins are crucial groups of PGRs regulating cell divisions and the inducting somatic embryogenesis (*Jiménez, 2001*; *Yang & Zhang, 2010*). Exogenously applied PGRs had a direct impact on endogenous PGRs abundance, which led to somatic embryogenesis induction. For example, the increase in endogenous auxin correlates with the embryonic response in *Quercus suber* (*Carneros et al., 2023*), *Cunninghamia lanceolata* (*Zhou et al., 2017*), and *Arabidopsis thaliana* (*Karami et al., 2023*), while changes in cytokinins abundance are responsible for the embryonic regeneration in *Coffea canephora* which is based on participating cytokinins like N6-(2-Isopentenyl)adenine (2iP), 6-benzyladenine (BA), or Kinetin (*Avilez-Montalvo et al., 2022*; *Hatanaka et al., 1991*). As crosstalk between PGRs occurs in the developmental process of SE, the ratio between endogenous PGRs can dictate the embryonic switches (*Asghar et al., 2023*), for instance, the cytokinin/Indole-3-acetic acid (IAA) ratio in *C. canephora*, the IAA/Abscisic acid (ABA) ratio in *Triticum aestivum* (*Hess & Carman, 1998*), the IAA/ABA or 2iP/Zeatin cytokinin ratios in *Corylus avellana* are an indication of embryogenic competence (*Centeno et al., 1997*), and IAA/gibberellic acid (GA3) ratio, which is an indication of the embryonic response among explants used in tissue culture in Rosa 'John F. Kennedy' (hybrid tea rose) (*Du et al., 2024*).

In the *Coffea* genus, five decades after SE was defined for propagating coffee plants (*Staritsky, 1970*), improvements have been reported for high frequency propagation (*Berthouly & Michaux-Ferriere, 1996*; *Etienne, 2005*; *Gatica-Arias, Arrieta-Espinoza & Espinoza Esquivel, 2008*; *Quiroz-Figueroa et al., 2006*). Within the *Coffea* genus, the diploid species *C. canephora* utilizes gametophytic S-RNAse–based self-incompatibility, thus making the use of *in vitro* propagation useful to increase the homogeneity of the crop (*Nowak et al., 2011*). Besides pruning coffee trees, diseased or dead coffee trees replacement is required to keep the yield of coffee berries along the time in the field.

We have used an SE protocol for *C. canephora* that completes the induction and expression of SE in two periods of *in vitro* culture in tandem. Cotyledonary somatic embryos produced by this SE protocol can germinate in PGR-free MS media. The molecular and cellular mechanisms implicated in converting the somatic cell into the somatic embryo are not completely known. Involving the YUCCA-dependent auxin synthesis, auxin response factors (ARFs) and auxin (Aux)/IAA-dependent signaling, and PIN-FORMED (PIN)-mediated transport of auxin in SE of *C. canephora* has been revealed through biochemical and RNA-seq studies (*Quintana-Escobar et al., 2019*; *Uc-Chuc et al., 2020*). Modulation of proteins related to citokinin activation and distribution of auxins, as well as the dynamic of protein abundance during the SE of *C. canephora* are emerging (*Quintana-Escobar et al., 2023*), but proteins responsible for the reprogramming embryogenic response are still waiting to be determined. Here, we focus on determining the protein profile in leaves that serve as cut-disk explants, comparing proteins from leaves

of *in vitro* plant regenerants priming with or without PGR, referred to as +NAA–KIN and −NAA–KIN, respectively.

## MATERIALS AND METHODS

### Plant material and somatic embryos quantification

Pretreatment of individual *in vitro* 9-months old plantlets with six pairs of leaves was followed by induction cut-leaf explants to promote the somatic embryogenesis at the cut-leaf edges of *C. canephora*, as previously described with the following modifications (*Quiroz-Figueroa et al., 2006*). *In vitro* plants were maintained under a long-day photoperiod in Murashige and Skoog (MS) media until they were 9-months old, then transferred to supplemented MS media with 0.54 µM 1-naphthaleneacetic acid (NAA) and 2.32 µM kinetin (KIN) for 14 days, which are referred to as +NAA–KIN plantlets, or retained in MS without any PGR, which are referred to as −NAA–KIN plantlets. Then 0.8-cm circular explants from the third to the fourth mature leaf of five plants from each treatment, either +NAA–KIN or −NAA–KIN, were cultivated in liquid Yasuda medium supplemented with 5 µM 6-benzyladenine (BA) in darkness for up to 100 days. Somatic embryos derived from +NAA–KIN and −NAA–KIN explants were examined and quantified under a stereomicroscope (SMZ745T; Nikon, Tokyo, Japan) with an adapted camera (EOS Rebel T3i; Canon, Melville, NY, USA) at magnification of 10× for up to 100 days after induction of cut-leaf explants in the Yasuda-based media (*Yasuda, Fujii & Yamaguchi, 1985*). The Student paired-samples *t*-test in GraphPad prism 9 version 9.0.1 static analysis software was used to determine significant differences between treatments. Differences were considered statistically significant with values of $P \leq 0.0001$.

### Protein sample preparation for two-dimensional sodium dodecyl sulfate-polyacrylamide gel electrophoresis

Nine-month-old plantlets were transferred to MS medium either with or without NAA-KIN, kept for 14 days, and then the leaf explants were dissected and frozen for protein analysis. The frozen leaf powder was homogenized in an extraction buffer that contained 0.7 M sucrose, 100 mM KCl, 500 mM Tris-Cl (pH 7.5), 50 mM EDTA, 50 mM dithiothreitol (DTT), 1 mM phenyl methyl sulfonyl fluoride, and a cocktail of protease inhibitors under icy conditions (*Hurkman & Tanaka, 1986*). All steps were performed on ice and with ice-cold solutions. After centrifuging the homogenate at 15,000 *g* for 10 min, the supernatant was made 10% trichloroacetic acid/50% acetone and 5 mM DTT and incubated on ice for 5 min (*Niu et al., 2018*). The protein precipitate obtained by centrifugation at 15,000 *g* for 3 min was further washed twice with 80% acetone supplemented with 5 mM DTT. When the residual acetone in samples was removed by air drying, the protein precipitate was dissolved in rehydration buffer consisting of 8 M urea, 1.5 M thiourea, 1.5% CHAPS, and 50 mM DTT. The protein amount in the samples was determined by the Peterson method (*Peterson, 1977*).

## Two-dimensional sodium dodecyl sulfate-polyacrylamide gel electrophoresis and gel image analysis

Approximately 900 µg of protein was loaded on a 24-cm IPG-strip with a linear pH range between 3 and 10 (ReadyStrip IPG; Bio-Rad, Hercules, CA, USA) by 12-h passive in-gel rehydration. Isoelectric focusing was performed at 20 °C and initiated with a linear gradient to 100 V over 4 h, followed by 250 V for 4 h, 1,000 V for 1-h, linear gradient to 10,000 over 2 h, 10,000 V for 80,000 V-h, and 200 V for 1-h. The gel strips were then equilibrated in 5 mL of the equilibrated solution I (6 M urea, 375 mM Tris-HCl, pH 8.8, 2% sodium dodecyl sulfate (SDS), 20% glycerol, and 2% DTT), followed by an equilibration in 5 mL of a solution supplemented with 2.5% iodoacetamide without DTT. For SDS-PAGE, equilibrated gel strips were placed on top of 12% gel and overlaid with 1% (wt/vol) agarose in equilibration solution I. Next, the electrophoresis was performed at 12 °C with 200 V until the dye reached the bottom of the gel. The gels (25 × 20.5 × 1.5 mm) were run in the Protean plus Dodeca Cell System (Bio-Rad, Hercules, California, USA). After electrophoresis, the gels were rinsed three times with Milli-Q water, stained for 1-h with G-250 staining (PageBlue; Thermo Fisher Scientific, Waltham, MA, USA), and rinsed with Milli-Q water to remove the background. Three biological replicates were made. Digital images of gels were acquired at 300 dpi using the ChemiDoc MP System (Bio-Rad, Hercules, California, USA). Comparison of gel images and processing, and quantitative comparison of protein spots based on their percent volume were performed with Melanie version 8 (GE Healthcare, Chicago, IL, USA). Significant changes in protein abundance between +NAA−KIN and −NAA−KIN samples were found with one-way analysis-of-variance test at $P < 0.05$ and at least 1.4-fold change using Melanie software.

## Protein in-gel digestion and identification by liquid chromatography tandem mass spectrometry

Spots with statistical significance were manually excised and subjected to in-gel digestion. Tryptic peptides were desalted with ziptip tips, as described in *Huerta-Ocampo et al. (2012)*. The peptides were resuspended in 0.1% formic acid and analyzed using the Easy-nLC-1000 liquid chromatography system (Thermo Fisher Scientific, Waltham, MA, USA) with the analytical nanoscale liquid chromatography column PepMAP (Thermo Fisher Scientific, Waltham, MA, USA). Peptide elution was made with a binary gradient of water as mobile phase A and acetonitrile as mobile phase B supplemented with 0.1% formic acid and delivered at 250 µL min$^{-1}$ using the following program: 5% of B to 55% B in 35 min, 55% B to 100% B in 5 min, 100% B for 5 min, 100% B to 5% B in 5 min followed by column equilibration with 5% B for 7 min. Mass spectrometry was performed using the LTQ-Orbitrap mass spectrometer (Thermo Fisher Scientific, Waltham, MA, USA) operated in positive ion mode and with an easy-nanospray potential of 1.9 kV. Data-dependent acquisition was performed at Orbitrap for the top-10 method with 120,000 resolution for full MS and an AGC target of 1,000,000 and 15,000 resolution and AGC target of 50,000 for tandem mass spectrometry (MS/MS). The ions were fragmented in the higher energy dissociation cell (HCD). The ion threshold for triggering MS/MS events was 5,000, and dynamic exclusion was 90 s. Proteome Discovery 2.2 (Thermo Fisher

Scientific, Waltham, MA, USA) was used for data processing using MASCOT (Matrix Science Inc., Boston, MA, USA) and SEQUEST as search engines and the *Coffea canephora* protein sequence database (GCA_900059795.1) from the National Center for Biotechnology Information. The search engine parameters were as follows: carbamidomethylating of cysteine as a fixed modification, oxidation of methionine as a variable modification, trypsin enzyme, maximum missed cleavage 2, and *Coffea* taxonomy. We used a decoy database and only recovered protein hits based on two successful peptide identifications. The mass spectrometry proteomics data have been deposited to the ProteomeXchange Consortium *via* the PRIDE (*Perez-Riverol et al., 2022*) partner repository with the dataset identifier PXD055039.

## Western blot analysis

Nine-month-old plantlets were transferred to MS medium either with or without NAA–KIN and kept for 14 days. Equivalent to 100 mg of leaf explantes from +NAA–KIN and −NAA–KIN plantlets were frozen in liquid nitrogen, ground with a pestle and mortar, homogenized in Laemmli buffer (*Laemmli, 1970*). Proteins were resolved onto 12% SDS-polyacrylamide gel and transferred to a polyvinylidene difluoride (PVDF) membrane (Immobilon-P; Merck Millipore). The membrane was blocked with 5% bovine serum albumin (BSA) in TBS or 5% fat-free milk in PBS for 1 h at room temperature and then probed with anti-Ub (dilution 1:5000; ab19169; Abcam, Cambridge, MA, USA) or with anti-HIS3 (dilution 1:10,000) as loading control (ab1791; Abcam, Cambridge, MA, USA), respectively (*Aguilar-Hernández et al., 2017*). AP conjugated antibody was utilized to detect immuno complexes by using 1-step NBT/BCIP regents (Thermo Fisher Scientific, Rockford, IL, USA).

## Sequence analysis and primer design for *C. canephora*

Given that two *RPN12* genes were reported in *Arabidopsis thaliana AtRPN12A* (AT1G64520.1) and *AtRPN12B* (AT5G42040.1), and that *AtACT2* (AT3G18780.2) is a reference gene for quantitative real-time PCR (RT-qPCR) (*Book et al., 2010*; *Zhou et al., 2019*), orthologs of those genes in *C. canephora* were retrieved in BLAST searches against *C. canephora* implemented in Coffee Genome Hub resource (http://www.coffee-genome. org). *C. canephora* encodes only one *CcRPN12* gene (Cc08t16370.1), with 75.13% and 75.84% of identity to *A. thaliana AtRPN12A* and *AtRPN12B*, respectively. The homolog of *A. thaliana AtACT2* in *C. canephora* was identified as Cc07t17400.1 (*CcACT2*). Specific primers for RT-qPCR analysis of *CcRNP12* and *CcACT2* in *C. canephora* were designed from the coding sequences (CDS) using the software Primer 3 (https://primer3.ut.ee). The primers were synthesized by Adn-Artifitial (Guanajuato, Mexico) on a scale of 25 nmol, with the option of desalted purification, without modifications in the 5′ and 3′ ends. The amplicon of the *CcRPN12* gene spans exons one to three and has a length of 249 bp. The *CcACT2* gene amplicon was 233 bp in length.

## RNA extraction, cDNA synthesis and quantitative real-time PCR (RT-qPCR) analysis

Nine-month-old plantlets were transferred to MS medium either with or without NAA–KIN and kept for 14 days. An equivalent of 100 mg of leaf explants was collected, frozen with liquid nitrogen, and preserved in the Thermo Scientific™ Revco ExF Series Ultra Freezer for 30 days. The total RNA was extracted using TRIzol reagent (Invitrogen, Waltham, Massachusetts, U.S.), in accordance with previous reports by *Chomczynski (1993)*. RNA integrity was assessed by examining samples in a 1.2% agarose gel, and by measuring both A260/A280 and A260/A230 ratios using a NanoDrop 2000 spectrophotometer (Thermo Fisher Scientific, Waltham, MA, USA). The samples used had measurements of A260/A280 ratio between 1.89 and 1.99. The RNA concentrations of the processed samples ranged from 240.4 to 353.4 ng/μL. One μg of RNA was used to synthesize cDNA with 200 units of the ImProm-II™ reverse transcriptase (Cat # A3802; Promega, Madison, WI, USA) and 0.5 μg oligo $dT_{18}$ primers (Cat # SO131; Thermo Fisher Scientific, Waltham, MA, USA), as indicated by the manufacturer's instructions, in a volume of 20 μL.

The *C. canephora* ACT2-specific primers CcACT2Fwd (5′-AGCAACTGGGATGACATGGA) and CcACTRev (5′-TCCAGCACAATACCAGTCGT), and the RPN12-specific primers CcRPN12Fwd (5′-ATTCAAAGCTGCCTTCGTCC) and CcRPN12Rev (5′-CCTCGGGCATCAGTGTAGTA) were utilized in the RT-PCR reaction setup. For each individual PCR reaction, 100 ng of cDNA was utilized as a template for PCR amplification in a 20 μL reaction volume using 10 μL of QuantiNova Probe PCR Master Mix (Cat # 208054; QIAGEN, Hilden, Germany), 1 μL each of 8 μM forward and reverse primers, and Rotor Gene Q MDx instrument (QIAGEN, Venlo, The Netherlands). Three technical replicates were made for each reaction with Rep. Ct (95% CI). The reaction parameters were 95 °C for 2 min, 40 cycles of 95 °C for 15 s and 61 °C for 30 s. Fluorescence data were collected during each annealing-extension step at 61 °C. Five dilution points were used for the standard curve 1, 1:10, 1:100, 1:1,000, and 1:10,000. For the *ACT2* gene pair of primers, the efficiency was 141.18%, R2 was 0.8922, and the slope was −2.6155. For the *RPN12* gene primer pair, the efficiency was 85.78%, the R2 value was 0.9914, and the slope was −3.7175. The RT-qPCR data reactions that were used to calibrate curves, melt, identify outliers, reproducibility (Rep. Ct (95% CI)), and variation were first evaluated using the Rotor-Gene Q 2.1.0.9 Windows software version, and then data were extracted for use in Excel templates. Subsequently, the relative transcript abundance of the *RPN12* gene was determined using the Delta-Delta Ct ($2^{-\Delta\Delta Ct}$) method (*Livak & Schmittgen, 2001*), with the *ACT2* reference gene as internal control (*Gutierrez et al., 2008*). The significant changes in *RPN12* gene expression in the +NAA–KIN and −NAA–KIN explants were determined using a Student's *t*-test. GraphPad Prism 9 version 9.0.1 statistical analysis software was employed to analyze gene expression data with statistically significant values of $P \leq 0.05$.

## Histology

To carry out histological and histochemical analysis on +NAA–KIN and −NAA–KIN explants, the samples were immersed in FAA solution (containing 10% formaldehyde, 5% acetic acid, and 50% ethanol) for 48 h before washing five times with distilled water. Next, the samples were dehydrated in a graded ethanol series starting with 10%, followed by 30%, 50%, 70%, 85%, 96%, and 100%. The steps were repeated, and the samples were vacuum infiltrated for 20 min, followed by an hour of incubation at 4 °C. After that, the samples were coated with JB-4 resin (JB-4 Embedding kit; Polysciences, Warrington, PA, USA). The blocks were cut into 3 μm slices using a MICROM HM 325 microtome and were double-stained with a solution of toluidine blue to characterize their structural features and with Xylidine Ponceau to detect their total protein (XP, Vidal 1970). Images were acquired using a Leica MZFL III stereomicroscope.

## RESULTS

### Conditioning explants with NAA and KIN

We previously defined the auxin plus cytokinin priming plantlet step to promote forming somatic embryos along cut-leaf edges of *C. canephora* explants. The *in vitro* plantlets were given 0.54 μM auxin 1-naphtaleneacetic acid (NAA) plus 2.32 μM cytokinin kinetin (KIN) in semisolid culture media for 14 days, which is referred to as +NAA–KIN. The next step was to give 5 μM cytokinin 6-benzyladenine (BA) to cut-leaf explants in liquid media for up to 100 days (*Quiroz-Figueroa et al., 2006*). The aim of our analysis was to discover how NAA plus KIN affect the embryogenic response of this SE setting. Plantlets were cultivated for 14 days either with or without NAA–KIN which are referred to as +NAA–KIN and −NAA–KIN, respectively, and then used as cut-leaf explants to induce SE through BA. NAA–KIN treatment did not affect the morpho-anatomical patterns of *C. canephora* plant leaves during a 14-day period (Fig. S1). Compared with +NAA–KIN explants, −NAA–KIN explants showed less growth of meristematic-like cells at the edge, and proembryonary masses were observed at 28 days of explant cultivation, as shown in Fig. S2. Next, embryos produced along the edge of the explant were recorded for up to 100 days (Fig. 1A). NAA–KIN further increased the globular stage embryo yield by two-fold (128/64) after 75 days of explant cultivation (Fig. 1B). After another 25 days of explant cultivation, the yield of embryos at the cotyledonary stage increased to 31 times (31/1). When comparing the number of embryos produced after 75 and 100 days of induction, it was found that the number of embryos at the globular stage in +NAA–KIN explants decreased dramatically while it remained constant in −NAA–KIN explants. The significant increase in the number of cotyledonary embryos in +NAA–KIN explants over −NAA–KIN explants suggests that NAA–KIN is involved in explant conditioning to improve SE.

### Two-dimensional sodium dodecyl sulfate-polyacrylamide gel electrophoresis (2D SDS-PAGE) analysis

Next, we followed to compare the protein patterns of +NAA–KIN and −NAA–KIN explants using two-dimensional polyacrylamide gel electrophoresis (2D SDS-PAGE). Multiple 2D SDS-PAGE gels were obtained to represent three technical and biological

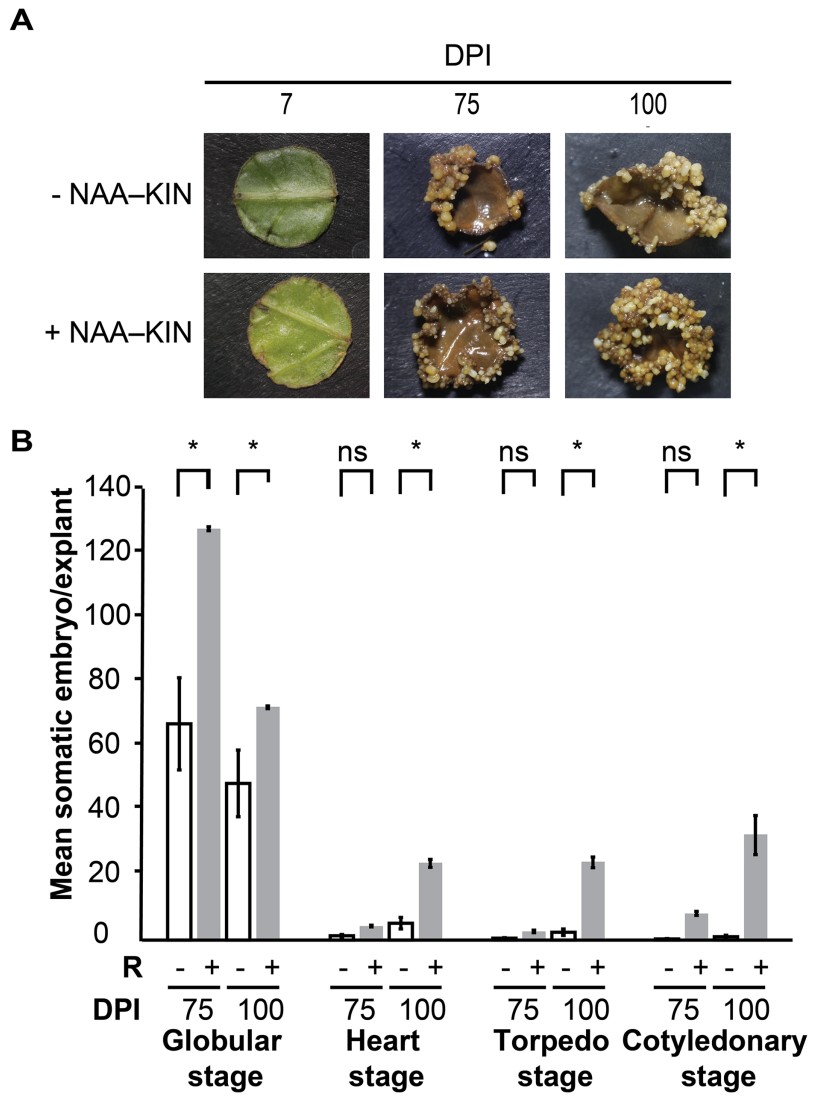

**Figure 1 Priming of plantlets with NAA plus KIN is required for a high yield of the somatic embryos in *C. canephora*.** (A) Representative cut-leaf explants from −NAA–KIN and +NAA–KIN plantlets after 7-, 75-, and 100-days of cultivation in Yasuda liquid media with BA at darkness. DPI indicates days post induction. (B) Quantification of somatic embryos in cut-leaf explants from −NAA–KIN and +NAA–KIN plantlets after 7-, 75-, and 100-days of cultivation in Yasuda liquid media with BA at darkness. White bars indicate somatic embryos produced with −NAA–KIN explants, while black bars indicate somatic embryos produced with +NAA–KIN explants. Each bar represents the mean number of at least four explants from three independent replicates. Error represents the standard error. Asterisk indicates a significant difference with $P < 0.0001$.

repeats. The reference map contained 230 protein spots and was populated between pI from 4 to 7 and from 50- to 100 kDa MW. Forty-two differential spots were identified with at least a 1.4-fold change and a *P*-value of 0.05 in the one-way analysis of variance test. Forty-one spots were differentially up-accumulated, and one spot was down-accumulated (Fig. 2, Fig. S3). The top five protein spots with the most abundance ratios were 1, 3, 4, 2 and 7. Twelve out 42 2D-PAGE differential spots were identified with liquid chromatography MS/MS using the *C. canephora* genome dataset deposited at the National

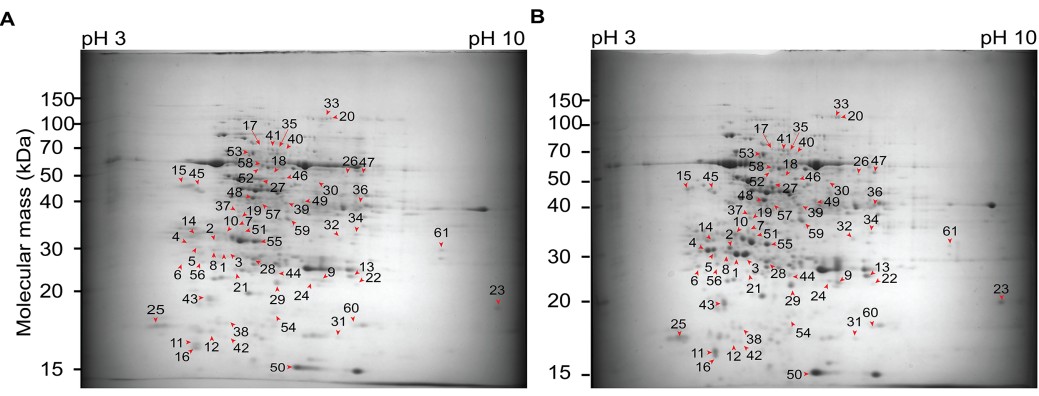

**Figure 2 Two-dimensional gel electrophoresis images and spot analysis revealed 42 proteins with differential abundance in cut-leaf explants before they were cultured in Yasuda liquid media at darkness.** (A) Representative gel images from cut-leaf −NAA–KIN explant samples. (B) Representative gel images from cut-leaf and +NAA–KIN explants samples. Differentially accumulated proteins were stained with colloidal Coomassie G-250. Red arrows indicate differentially accumulated protein spots.

Center for Biotechnology Information dataset. The ratio changes for spots 1 (CDP17851), spot 4 (CDP17853), and spot 2 (CDP17852), all of which are chitinases A class III, and spot 3 (CDP12187), which is a subunit RPN12 of the 26S proteasome, were more than five-fold (Table 1).

Proteins with a role in the protein metabolic process include a ribosomal protein CDP14047 (spot 37), leucine aminopeptidase CDP04332 (spot 17), papain-like protease CDP01294 (spot 7), and the subunit of the ATP-dependent proteolytic complex 26S proteasome CDP12187 (spot 3). Proteins participating in folding and refolding proteins include a chloroplast chaperonin CPN60 CDP18583 (spot 53) and a mitochondrial heat shock protein 60 CDO99385 (spot 53). Phosphatase protein is implicated in the reversible phosphorylation of proteins CDP17727 (spot 7) and in chromatin organization by recognition of methylated dinucleotides CpG as well CDP05220 (spot 46). Proteins associated with oxidation–reduction were NADPH-dependent thioredoxin reductase CDP01751 (spot 59) and isoflavone reductase CDP07496 (spot 59), glycine catabolic process CDP08258 (spot 17), photosynthesis CDP20139 and CDP17859 (spots 50 and 7), and carbohydrate metabolic process CDP17851, CDP17853, and CDP17852 (spots 1, 4, and 2).

## RPN12 gene expression analysis and western blot analysis of extracts from −NAA–KIN or +NAA–KIN explants with ubiquitin antibody

The embryogenic response and altering proteome constituents through activating protein biosynthesis, post-translational modifications of proteins or removal of unnecessary proteins have been correlated by biochemical and morphological analysis (*Aguilar-Hernández & Loyola-Vargas, 2018*; *Gomez-Garay et al., 2013*; *Kumaravel et al., 2017*). The multicatalytic ATP-dependent complex 26S proteasome is responsible for specific protein degradation after proteins are marked with ubiquitin (*Vierstra, 2009*). In this study, the protein subunit CcRPN12 of the ATP-dependent proteolytic complex 26S proteasome was

**Table 1 List of proteins with differential abundance identified by LC-MS/MS.**

| Spot | Protein function | Genbak/Coffe ID | Theoretical Mr/pI | Experimental Mr/pI | Anova (p) | Sequence coverage (%) | PSMs | Score mascot | Score sequest HT | Ratio +NAA−KIN/− NAA−KIN |
|---|---|---|---|---|---|---|---|---|---|---|
| **Carbohydrate metabolic process** | | | | | | | | | | |
| 1 | Chitinase A class III (CHIA/LYS1) | CDP17851/ Cc05_g00760 | 35.6/5.86 | 29.8/5.20 | 0.009 | 6 | 18 | 119 | 1.96 | 11.5 |
| 4 | Chitinase A class III (CHIA/LYS1/SE2) | CDP17853/ Cc05_g00780 | 34.9/5.03 | 30.6/4.75 | 3.91E−05 | 7 | 10 | 157 | 10.43 | 7.6 |
| 2 | Chitinase A class III (CHIA/LYS1) | CDP17852/ Cc05_g00770 | 35.9/5.35 | 31.2/5.11 | 4.73E−04 | 15 | 29 | 214 | 15.8 | 7.5 |
| **Protein metabolic process** | | | | | | | | | | |
| 3 | Regulatory particle non-atpase 12a (RPN12A) | CDP12187/ Cc08_g16370 | 30.8/5.24 | 29.8/5.34 | 1.32E−06 | 12 | 5 | 48 | 0 | 8.5 |
| 7 | Responsive to dehydration 21a (RD21A) | CDP01294/ Cc10_g04820 | 51.4/5.16 | 34.8/5.38 | 1.12E−04 | 14 | 10 | 50 | 7.37 | 2.5 |
| 7 | Probable protein phosphatase 2C 71 (WIN2) | CDP17727/ Cc00_g02680 | 31.5/5.16 | 34.8/5.38 | 1.12E−04 | 7 | 6 | 128 | 6.65 | 2.5 |
| **Photosynthesis** | | | | | | | | | | |
| 50 | Ribulose bisphosphate carboxylase small chain (SSU11) | CDP20139/ Cc00_g15710 | 18.2/8.22 | 15.1/6.40 | 0.040 | 32 | 287 | 1,288 | 71.55 | 1.9 |
| 7 | Photosystem II subunit p, photosystem ii subunit p-1 (PSII-P) | CDP17859/ Cc05_g00840 | 28.5/8.1 | 34.8/5.38 | 0.012 | 17 | 13 | 67 | 3.54 | 2.5 |
| **Glycine catabolic process** | | | | | | | | | | |
| 17 | Glycine decarboxylase p-protein 2 (GLDP2) | CDP08258/ Cc08_g10590 | 112.7/7.8 | 72.4/5.74 | 0.006 | 8 | 23 | 168 | 10.71 | 1.8 |
| 17 | Putative Leukotriene A-4 hydrolase homolog | CDP04332/ Cc09_g02360 | 69.2/5.54 | 72.4/5.74 | 0.006 | 13 | 48 | 581 | 26.17 | 1.8 |
| **Ribosome** | | | | | | | | | | |
| 37 | 60S acidic ribosomal protein P0(RPP0A) | CDP14047/ Cc02_g06780 | 34.3/5.1 | 38.8/5.39 | 3.56E−04 | 13 | 19 | 84 | 3.84 | 1.7 |
| **Refolding and folding protein** | | | | | | | | | | |
| 53 | Chaperonin-60beta2 (CPN60β2) | CDP18583/ Cc01_g00200 | 64.6/5.77 | 67.2/5.61 | 0.033 | 19 | 33 | 252 | 25.1 | 1.6 |
| 53 | Heat shock protein 60-3b (HSP60-3B) | CDO99385/ Cc03_g07040 | 61.1/5.76 | 67.2/5.61 | 0.010 | 11 | 11 | 118 | 9.58 | 1.6 |

(Continued)

| Spot | Protein function | Genbak/Coffe ID | Theoretical Mr/pI | Experimental Mr/pI | Anova (p) | Sequence coverage (%) | PSMs | Score mascot | Score sequest HT | Ratio +NAA–KIN/– NAA–KIN |
|------|------------------|-----------------|-------------------|--------------------|-----------|-----------------------|------|--------------|------------------|--------------------------|
| **Antioxidant activity** | | | | | | | | | | |
| 59 | NADPH-dependent thioredoxin reductase a (NTR2/ NTRA) | CDP01751/ Cc07_g12230 | 34.9/6.24 | 37.0/6.18 | 0.003 | 9 | 6 | 55 | 5.41 | 1.5 |
| 59 | Isoflavone reductase | CDP07496/ Cc10_g02660 | 33.7/7.06 | 37.0/6.18 | 0.003 | 11 | 7 | 71 | 2.35 | 1.5 |
| **Chromatin organization** | | | | | | | | | | |
| 46 | Methyl-cpg-binding domain protein 13 (MBD13) | CDP05220/ Cc02_g03140 | 52.9/6.35 | 49.9/6.09 | 0.016 | 11 | 3 | 0 | NA | 1.5 |
| **Unknown** | | | | | | | | | | |
| 7 | Putative N-acylneuraminate-9-phosphatase (NAMP) | CDP09530/ Cc06_g18960 | 34.2/6.32 | 34.8/5.38 | 0.012 | 6 | 4 | 113 | 3.42 | 2.5 |
| 23 | Citrate-binding protein-like | CDP12834/ Cc07_g16910 | 24.1/9.29 | 20.5/9.18 | 0.029 | 19 | 7 | 126 | 5.89 | 1.7 |

found to be up-regulated. Then, we opted to investigate if the rise in CcRPN12 protein abundance in +NAA–KIN explants was determined by the transcript-level using qPCR with the *CcACT2* gene as a reference. There was no difference in the amount of *CcRPN12* in +NAA–KIN or −NAA–KIN explants (Fig. 3A), suggesting a post-transcriptional mechanism leading the up-regulation of CcRNP12 in +NAA–KIN explants. The RPN12 protein is essential for the proteasome assembly (*Boussardon et al., 2022*; *Smalle et al., 2002*), thus altering CcRPN12 abundance might impact the pool of ubiquitinated proteins. One of the overall ways to visualize ubiquitination can be achieved by western blotting against ubiquitin (*Aguilar-Hernández et al., 2017*). Given that within identified proteins in this study are implicated in the proteasome 26S dependent protein catabolic process, we determined the ubiquitination smear in −NAA–KIN and +NAA–KIN explants by western blotting with anti-ubiquitin and using anti-histone 3 as loading control. Both −NAA–KIN and +NAA–KIN explants displayed a similar ubiquitination smear at the high MW (Fig. 3B), suggesting that the 26S proteasome is active and that probably a small pool of ubiquitin conjugates engages in the totipotency of explants to produce efficiently somatic embryos.

## DISCUSSION

Somatic embryo proliferation is largely affected by the *in vitro* culture settings and plant species, as demonstrated by the variety of SE settings in the *Coffea* genus (*Aguilar et al., 2022*; *Loyola-Vargas et al., 2016*). Recent efforts to elucidate factors driving somatic

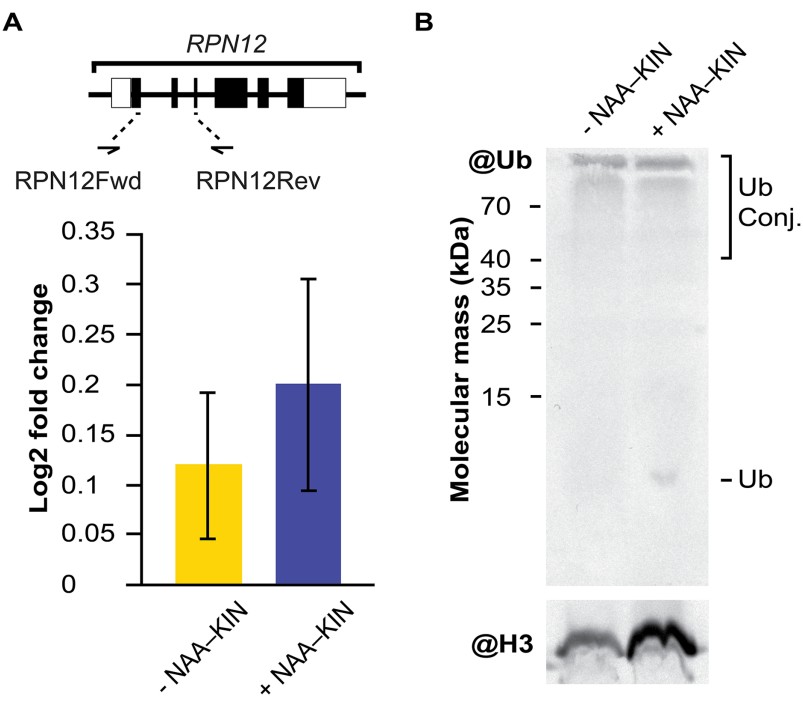

**Figure 3 The transcript abundance of *CcRPN12* and ubiquitin conjugates in −NAA–KIN and +NAA–KIN plantlets.** (A) Diagram of the *CcRPN12* gene and its expression pattern using RT-qPCR. Lines denote introns and boxes denote exons. Log2-fold change values were determined by RT-qPCR for *CcRPN12* in cut-leaf explants from −NAA–KIN and +NAA–KIN plantlets before cultivation in Yasuda liquid media at dark. (B) Ubiquitin conjugates in cut-leaf explants from −NAA–KIN and +NAA–KIN plantlets before cultivation in Yasuda liquid media at darkness. Ubiquitin monomer and conjugates are indicated. Antibody against histone 3 (H3) was used as loading control.

embryo proliferation at the edges of explants have led to the discovery implicating of the *de novo* auxin synthesis and ARF-mediated signaling (*Uc-Chuc et al., 2020*). This work has confirmed that an adequate explant with auxin plus cytokinin leads to efficient induction of SE on the cut-leaf edge of explants through BA and reported proteins involved in metabolism and the folding of proteins, photosynthesis, antioxidant activity, and transcriptional regulation. Validating the abundances of identified proteins in overlapped spots will require further investigation (*Hunsucker & Duncan, 2006*; *Pietrogrande et al., 2003*), such as using narrow IPG gels or shotgun proteomics approaches.

To study the impact of priming leaf explants with auxin plus cytokinin for 14 days before inducing SE through BA, this work used a protocol developed in *C. canephora* to achieve direct SE (*Quiroz-Figueroa et al., 2006*). The SE protocol is based on cut-leaf explants, where embryogenic responses through BA were produced along the edges of the cut-leaf. Thus, when somatic embryogenesis was performed with two types of explants using either primed explants with two PGRs (0.54 µM auxin 1-naphtaleneacetic acid (NAA) + 2.32 µM cytokinin kinetin (KIN)) for 14 days or in PGRfree for 14 days as control, the impact of priming leaf explants with PGRs on the embryogenic response through BA has been revealed. Plantlets cultivated in PGR-free (−NAA–KIN) media as explant sources showed significantly weaker totipotency than those cultivated with auxin

plus cytokinin (+NAA–KIN) as less embryos were recorded. Our findings confirm that under proper *in vitro* culture settings, proliferation of somatic embryos might occur (*Bidabadi & Jain, 2020*; *Quiroz-Figueroa et al., 2002*; *Yasuda, Fujii & Yamaguchi, 1985*). The importance of PGRs balance for expression of SE is well documented. In particular, an endogenous increase in auxin and cytokinin abundance have been reported when plantlets are being primed with +NAA–KIN (*Avilez-Montalvo et al., 2022*; *Ayil-Gutiérrez et al., 2013*), thus the transient change in the ratio of cytokinin/auxin mediated by priming with +NAA–KIN contributes to the embryogenic response. PGR balance for expression of SE has also been observed in *Medicago falcata* (*Ivanova et al., 1994*). The developmental plasticity, which leads forming the embryo by direct somatic embryogenesis, has been linked not only to the PGR but also to the presence of redox regulators, the use of epigenetic inhibitors, small peptides and amino acids such as L-proline, amino sugars, and the setting temperature for cultivation (*de Almeida et al., 2014*; *Couillerot et al., 2012*; *Fehér, 2008*, *2015*; *Guo et al., 2024*). Evidence of synergic action of PGRs in induction of SE in *C. arabica* has been observed when 24-epibrassinolid (24-epiBR) with 2iP were utilized in the SE (*Chone et al., 2018*).

Given that proliferation of somatic embryos occurs in both −NAA–KIN and +NAA–KIN cut-leaf explants, most likely multiple embryogenic cell types exhibiting specific proliferative responses might be present in the explants. It is time to speculate that one embryogenic cell type undergoes somatic proliferation in a cytokinin-dependent fashion occurring in −NAA–KIN explants with BA, while the other requires conditioning by auxin plus cytokinin that occurs in +NAA–KIN explants through BA. Evidence suggests that adequate plantlets with auxin plus cytokinin facilitates the unicellular proliferation of somatic embryos along cut-leaf edges through BA (*Quiroz-Figueroa et al., 2002*). In *in vitro* culture, the embryogenic-specific response in *C. canephora* is partially explained by perceiving the exogenous auxin and cytokinin followed by the *de novo* synthesis of auxins and cytokinins and their transport under the photoperiod (*Ayil-Gutiérrez et al., 2013*; *Uc-Chuc et al., 2020*), and the negative effects of exogenous auxins on BA-based SE at darkness (*Hatanaka et al., 1991*; *Yasuda, Fujii & Yamaguchi, 1985*). Analysis of cut-leaf based SE in *C. canephora* with other cytokines besides BA, such as 2iP and kinetin, has shown that cytokines are absorbed in the edges of the cut-leaf but not transported into the leaf (*Hatanaka et al., 1991*). Knowledge on regulating endogenous molecular determinants in line with somatic embryo proliferation is emerging in species such as *Cyathea delgadii* Sternb (*Mikula et al., 2021*).

Auxin and cytokinin are master compounds regulators of growth and development. Auxin is perceived by the transport inhibitor response1/auxin F-Box (TIR1/AFB) E3 ubiquitin ligases and promotes the interaction of TIR1/AFBs with ubiquitination substrates such as the transcriptional repressors auxin/indole-3-acetic acid (Aux/IAA) proteins involved in auxin signaling (*Blázquez, Nelson & Weijers, 2020*; *Gray et al., 2001*; *Tan et al., 2007*). Cytokinin is perceived by histidine protein kinases (*Arabidopsis* Histidine Kinase) that perform a phosphorylation cascade through two component system and induce the response by ARRs (*Arkhipov et al., 2019*; *Inoue et al., 2001*; *Werner & Schmülling, 2009*). We found the subunit of the protein complex proteasome 26S RPN12 as

up-accumulated in an auxin and cytokinin-dependent fashion. This regulatory particle subunit is essential for the integrity of the proteasome 26S and proper degradation of ubiquitinated proteins. In *A. thaliana*, the *rpn12* mutant displays phenotypes associated with hyposensitivity to auxin and cytokinin with stabilization of ubiquitinated proteins, enlargement, and proliferation of cells (*Kurepa et al., 2009*; *Smalle et al., 2002*). Based on this fact, an explanation for the enhanced proliferation of embryos by adequate plantlets with auxin plus cytokinin could be an increase in the sensitivity to the exogenous auxin and cytokinin by the recognizing and turnover of ubiquitinated proteins. Other 26S proteasome subunits that have displayed differential abundance during the somatic embryogenic response or embryo maturation include PAA1 in *Vigna unguiculata*, *Vitis vinifera*, and *Quercus suber* (*Gomez-Garay et al., 2013*; *Nogueira et al., 2007*; *Zhang et al., 2009*), PBA1 in *Vigna unguiculata*, *Pinus pinaster*, and in *Q. suber* (*Gomez-Garay et al., 2013*; *Morel et al., 2014b*), PBF1 in *P. pinaster* (*Morel et al., 2014a*, *2014b*), RPT4 in *Carica papaya* (*Botini et al., 2021*), RPT5 in *Cyclamen persicum*, *Musa* spp. AAA cv. Grand Naine, saffron, and H99 inbred maize (*Kumaravel et al., 2017*; *Lyngved et al., 2008*; *Sharifi et al., 2012*; *Sun et al., 2013*), RPT1 in *Cyathea delgadii* (*Domzalska et al., 2017*), RPT3 in *C. persicum* (*Rode et al., 2012*), RPT2 in *P. pinaster* (*Morel et al., 2014a*), an RPN9 in sugarcane (*Heringer et al., 2017*). Although transcriptomic and proteomic studies on SE have shown the subunits of the proteasome 26S are tightly regulated, the ubiquitinated proteins during SE are unknown. With comparable ubiquitination smear between −NAA–KIN and +NAA–KIN cut-leaf explants, one could speculate that a few key ubiquitination targets might be regulated.

Multiple levels of regulation are involved in somatic embryos proliferation. The molecular and cellular events associated with the embryogenic state, in part, can be inferred from the proteins identified here. The stress and defense-related protein chitinase A class III (Spot 1, 2, and 4) has been reported in the SE (*De Jong et al., 1992*). Chitinase proteins have been identified during SE induction (*De Jong et al., 1992*; *Liu, Yang & Shen, 2015*), which bound to cell wall Arabinogalactan-proteins (AGPs) and deliver oligosaccharides that may act as signaling molecules (*Domon et al., 2000*; *van Hengel et al., 2001*). Reactive oxygen species (ROS) homeostasis maintains the oxidoreductase-reducing power *via* the thioredoxin system (spot 59) (*Marty et al., 2009*; *Zagorchev et al., 2012*). ROS homeostasis related proteins have been identified in *Gossypium hirsutum* (*Zhou et al., 2016*), *Elaeis guineensis* (*Aroonluk, Roytrakul & Jantasuriyarat, 2019*), and *Catharanthus roseus* (*Gulzar et al., 2019*) during SE. Chromatin organization is facilitated by recruiting DNA organizing proteins such as methyl CpG binding proteins (spot 46) and histone deacetylase protein complex, which is in agreement with histone methylation seen in the SE of *C. canephora* (*Grzybkowska, Nowak & Gaj, 2020*; *Nic-Can et al., 2013*). Free amino acid homeostasis *via* hydrolysis of proteins and amino acid leucine from the N-terminus end has been suggested for PEG-dependent SE in *Carica papaya* (*Bartos et al., 2018*; *Matsui, Fowler & Walling, 2006*; *Vale Ede et al., 2014*), and glycine directed to the biosynthesis of serine *via* glycine decarboxylation. This is in line with the findings of serine hydroxymethyltransferase, which is associated with the competence in sugarcane callus (*Xavier et al., 2022*).

## CONCLUSIONS

In summary, we have shown that plantlet conditioning with auxin plus cytokinin is required for the high yield of somatic embryogenesis in the protocol. An adequate plantlet with auxin plus cytokinin not only increases the proliferation of somatic embryo, but also switches up and down the accumulation of proteins associated with them. The proteins identified are involved in the metabolism and folding of proteins. The identified protein RPN12, which is a subunit of the regulatory particle of the 26S proteasome exhibits that ubiquitination could be a new layer in embryogenic competence regulation in *C. canephora*. One of the main challenges in the future is to isolate the ubiquitination substrates while priming explants and induction of somatic embryogenesis.

## ACKNOWLEDGEMENTS

We thank the French National Research Institute for Sustainable Development (IRD), French Agricultural Research Centre for International Development (CIRAD) and collaborators for publication access to the Coffee Genome Hub, and Carlos Oropeza for his technical assistance.

### Funding

Evelyn A. Carrillo-Bermejo received financial support from CONAHCYT #814960 PhD scholarship grant. The funders had no role in study design, data collection and analysis, decision to publish, or preparation of the manuscript.

### Grant Disclosures

The following grant information was disclosed by the authors:
CONAHCYT #814960 PhD scholarship grant.

### Competing Interests

The authors declare that they have no competing interests.

### Author Contributions

- Evelyn A. Carrillo-Bermejo performed the experiments, analyzed the data, prepared figures and/or tables, authored or reviewed drafts of the article, and approved the final draft.
- Ligia Brito-Argáez performed the experiments, analyzed the data, prepared figures and/or tables, authored or reviewed drafts of the article, and approved the final draft.
- Rosa M. Galaz-Ávalos performed the experiments, authored or reviewed drafts of the article, and approved the final draft.
- Felipe Barredo-Pool conceived and designed the experiments, authored or reviewed drafts of the article, and approved the final draft.
- Víctor M. Loyola-Vargas conceived and designed the experiments, analyzed the data, authored or reviewed drafts of the article, and approved the final draft.

- Victor Aguilar-Hernández conceived and designed the experiments, performed the experiments, analyzed the data, prepared figures and/or tables, authored or reviewed drafts of the article, writing–original draft, and approved the final draft.

## Data Availability

The protein identification data are available at ProteomeXchange: PXD055039.

## Supplemental Information

Supplemental information for this article can be found online at http://dx.doi.org/10.7717/peerj.18372#supplemental-information.

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
