# Peer review of "Protein profile changes during priming explants to embryogenic response in Coffea canephora: identification of the RPN12 proteasome subunit involved in the protein degradation"

_PeerJ, doi:10.7717/peerj.18372_

## Round 0.1 · original submission · Major Revisions

The lack of novelty itself is not a reason for rejection in PeerJ, but I agree with reviewer 1 that qRT-PCR should be better contextualized and I also agree with reviewer 2 that raw data must be available. In this sense, all comments from both reviewers should be assessed.

Reviewer 1 ·

Basic reporting

The manuscript submitted by Carrillo-Bermejo et al. provides an insightful investigation into the effects of auxin and cytokinin on the protein profile during somatic embryogenesis in Coffea canephora. The language is clear throughout the document. However, some improvements are recommended in order to this manuscript be considered for publication.

- Expanding the introduction to include more details on the specific roles of auxin and cytokinin in SE could provide a more robust justification for the study.
- The authors should address the improvement of embryogenic competence acquisition using these key regulators in this process. This is the big picture of the work.
- Lines 167-168: insert the link or accession number of the reference proteome used in the work.
- RT-qPCR section is not well structured. Consider rewrite.
- Delta-delta Ct should be overwritten in the equation.
- Figure S1: the bars should be organized correctly. Provide legends to the figure (what is “E”, “SC”, the meaning of color staining, the sections used?)
- Line 255: the sentence is about +NAA-KIN, but the authors didn’t indicate the treatment at the beginning of the sentence.

Experimental design

- Methods: Protein sampling preparation – it is not clear the time point of sampling for protein analysis.
- It’s not clear the criteria to choose the top five spots (and the authors presented only four spots in the text).
- 293-294: the authors used RT-qPCR not Western Blot, consider change the section title. As the authors presented a Western Blot result, this should be addressed in the methods section.
- Why the authors performed the RT-qPCR only of RPN12 gene? Why not explored other important players involved in somatic embryogenesis found in proteomic analysis? Also, why used a homolog of Arabidposis as housekeeping gene instead a normalizer from Coffea canephora used in other researches? During the text, we don’t have sure if the gene worked well as normalizer.

Validity of the findings

The discussion focuses only on the components that the authors validated, and does not bring any new information. The discussion should be improved to achieve the proposed objectives.

Reviewer 2 ·

Basic reporting

The article “Auxin and cytokinin cause protein profile alterations to embryogenic response in Coffea canephora” addresses the proteomic changes of - NAA–KIN and + NAA–KIN explants using 2D SDS-PAGE. The basis of the manuscript is of great interest to the somatic embryogenesis (SE) community. Nonetheless, the manuscript should be improved as well as English improvements are required.
The introduction contains relevant information about SE and C. canephora. However, there is a lack of information about the proteomic studies in SE.
The article structure is fine. The figures are of good quality. Authors must check the bars in supplementary figure 1.
There is no raw data available for proteomics analysis. The data can be submitted to some repository but this is not mandatory.
The article title refers to protein profile alterations in somatic embryogenesis caused by auxin and cytokinin. This is a generic title, there is no novelty in it. Besides, there are two points to be considered, first, both phytohormones are known to cause cell alterations. Second, authors have performed 2D-SDS-PAGE, and twelve out of forty-two differential spots were identified. This does not represent a protein profile. I suggest authors modify the generic title to a more specific one, for example, focusing on the RPN12 protein.

Experimental design

Although the research question is well defined (the work aimed to discover how NAA and KIN affect the embryogenic response of SE using a protein profile approach), I am wondering why authors chose 2D SDS-PAGE as the technique to obtain the protein profile as the number of identified proteins is very low.
Another critical issue, the authors presented western blot results but there is no information about this methodology in the methods section. It is mandatory to include all methodologies in Methods.
Authors must better describe the information in line 100.
Authors must clarify which leaves explants (14 days, 100 days) were used for protein extraction. This information is given for RNA extraction (line 195).
Statistical analysis is not clear. Line 144 indicates that a one-way analysis of variance test was performed. Is this information correct?

Validity of the findings

The manuscript reports interesting results but detailed explanations are missing, mostly in the discussion section. Overall the manuscript needs clarification.
See below some detailed comments:
Line 50-51: .....” intense reshaping of the proteome”, give examples.
Line 52: “Most of the proteome rearrangements can be attributed to changes in transcription”, please, give some examples about the proteome rearrangements.
Line 54 - 55: Please revise the sentence – “the signaling cascades that are interconnected regulate activating or repressing of gene arrays”.
Line 57: Please specify which significant impacts are caused by the mentioned transcription factor.
Line 64: promotor?

---

## Round 0.2 · Minor Revisions

Dear authors,

Both referees attest that major changes were made. There are a few remarks made by reviewer 2 that should be incorporated.

Reviewer 1 ·

Basic reporting

N/A

Experimental design

N/A

Validity of the findings

N/A

Additional comments

N/A

Reviewer 2 ·

Basic reporting

Authors have improved the quality of the overall manuscript, including the information provided and English language. I thank you all for this. The authors have followed the suggestions given by rewievers. The manuscript’s content is now more robust and clear than in the first version.

Experimental design

The suggested modification were made by the authors

Validity of the findings

The suggested modification were made by the authors.

Additional comments

I would like to give minor suggestions.
Line 40 – “Data are available via ProteomeXchange ....” information could be given in the last paragraph.
Line 125 – In the sentence “Modulation and dynamics of proteins during the SE of C. canephora are emerging”, it seems some information is missing. E.g. Studies about the modulation and .......
Line 327 – “P-value of at least 0.05 in the one-way”. I suggest to remove the “at least” to avoid readers misunderstanding.
Line 360 – change “argue” to “suggesting”.
Line 464 – Please, check the gramar.
Line 478 – The same as mentioned above.

---

## Round 0.3 · accepted · Accept

I have reviewed the revised manuscript and confirm that the authors have addressed the reviewers' comments. Although the original reviewers were not consulted, I have assessed the revisions and find the current version satisfactory. I believe the manuscript is now ready for publication.